# Systematic Review and Meta-Analysis: Recurrent Laryngeal Nerve Variants and Their Implication in Surgery and Neck Pathologies, Using the Anatomical Quality Assurance (AQUA) Checklist

**DOI:** 10.3390/life13051077

**Published:** 2023-04-24

**Authors:** Juan José Valenzuela-Fuenzalida, Vicente Baeza-Garrido, María Fernanda Navia-Ramírez, Carolina Cariseo-Ávila, Alejandro Bruna-Mejías, Álvaro Becerra-Farfan, Esteban Lopez, Mathias Orellana Donoso, Walter Loyola-Sepulveda

**Affiliations:** 1Department of Morphology and Function, Faculty of Health Sciences, Universidad de las Américas, Santiago 8370040, Chile; 2Departamento de Morfología, Facultad de Medicina, Universidad Andrés Bello, Santiago 8370186, Chile; 3Departamento de Ciencias y Geografía, Facultad de Ciencias Naturales y Exactas, Universidad de Playa Ancha, Valparaíso 2360072, Chile; 4Departamento de Ciencias Química y Biológicas, Facultad de Ciencias de la Salud, Universidad Bernardo O’Higgins, Santiago 8370874, Chile; 5Escuela de Medicina, Universidad Finis Terrae, Santiago 7501015, Chile; 6Kinesiology School, Faculty of Health Sciences, Universidad de las Américas, Santiago 8370040, Chile

**Keywords:** recurrent laryngeal nerve, non-recurrent laryngeal nerve, clinical anatomy, methodology anatomical, clinical neck

## Abstract

Introduction: The recurrent laryngeal nerve (RLN) is the structure responsible for sensory and motor innervation of the larynx, and it has been shown that its lesion due to a lack of surgical rigor led to alterations such as respiratory obstruction due to vocal cords paralysis and permanent phonation impairment. The objectives of this review were to know the variants of the RLN and its clinical relevance in the neck region. Methods: This review considered specific scientific articles that were written in Spanish or English and published between 1960 and 2022. A systematic search was carried out in the electronic databases MEDLINE, WOS, CINAHL, SCOPUS, SCIELO, and Latin American and Caribbean Center for Information on Health Sciences to compile the available literature on the subject to be treated and was enrolled in PROSPERO. The included articles were studies that had a sample of RLN dissections or imaging, intervention group to look for RLN variants, or the comparison of the non-recurrent laryngeal nerve (NRLN) variants, and finally, its clinical correlations. Review articles and letters to the editor were excluded. All included articles were evaluated through quality assessment and risk of bias analysis using the methodological quality assurance tool for anatomical studies (AQUA). The extracted data in the meta-analysis were interpreted to calculate the prevalence of the RLN variants and their comparison and the relationship between the RLN and NRLN. The heterogeneity degree between included studies was assessed. Results: The included studies that showed variants of the RLN included in this review were 41, a total of 29,218. For the statistical analysis of the prevalence of the RLN variant, a forest plot was performed with 15 studies that met the condition of having a prevalence of less than 100%. As a result, the prevalence was shown to be 12% (95% CI, SD 0.11 to 0.14). Limitations that were present in this review were the publication bias of the included studies, the probability of not having carried out the most sensitive and specific search, and finally, the authors’ personal inclinations in selecting the articles. Discussion: This meta-analysis can be considered based on an update of the prevalence of RLN variants, in addition to considering that the results show some clinical correlations such as intra-surgical complications and with some pathologies and aspects function of the vocal cords, which could be a guideline in management prior to surgery or of interest for the diagnostic.

## 1. Introduction 

The inadequate functioning of the thyroid gland due to various conditions leads, in many cases, to a surgical intervention known as thyroidectomy, which consists of the partial or total removal of the thyroid gland. Because this surgery is performed in a region with a high density of organs and neurovascular structures per body area, it has great complexity. Therefore, the surgical anatomy of the region must necessarily be considered. Therefore, one of the structures to be reviewed in a surgical procedure is the recurrent laryngeal nerve (RLN) and its respective variations due to the pathological consequences that nerve injury entails [1].

RLN is a nerve structure responsible for the sensory and motor innervation of the larynx. RLN provides somatic motor innervation to the intrinsic larynx’s muscles, except for the cricothyroid muscle. Anatomically, the RLN occurs bilaterally, with origins from the right and left vagus nerves. The right RLN has its origin in the root of the neck at the level of the right subclavian artery: it follows a course below the artery and then ascends through the tracheoesophageal groove and ends by entering the larynx through the inferior hiatus at the level of the inferior constrictor muscle of the pharynx. On the other hand, the left RLN crosses the aortic arch at the level of the superior mediastinum to then moves posteriorly to this vascular structure and, as with the right RLN, ascends through the tracheoesophageal groove until it enters the larynx through the hiatus posterior to the inferior constrictor muscle [2,3,4,5].

RLN presents different types of anatomical variants, including extra-laryngeal branches, a distorted position, intertwined branches with the inferior thyroid artery, the presence of the non-recurrent laryngeal nerve (NRLN), NRLN surrounded by thyroid tissue from Zuckerkandl’s tubercle and a long distance in the extra-laryngeal bifurcation to the cricothyroid joint [6,7,8]. Extra-laryngeal ramifications of the RLN are characterized by bilateral bifurcations or trifurcations that occur before the nerve enters the larynx. These variations are commonly observed on the right side; in addition, the branches can also present as anterior and posterior ramifications at the level of the suspensory ligament of the thyroid gland (Berry’s ligament) [9,10].

The distortion of the RLN implies an abnormal orientation of its course, and this distortion is observed in patients with goiter or abnormal growth of the thyroid gland [11]. The intertwining between the branches of the RLN and the inferior thyroid artery is a variation that occurs in surgical procedures. This intertwining is made complex due to the presence of abundant branches of the inferior thyroid artery that make it difficult for one to identify the nerve [12].

NRLN, a rare variation of RLN, is frequently seen on the right side. NRLN is classified into three types, namely type I, II, and III, each of which has three subtypes. Type I corresponds to a cephalic course with respect to the bifurcation of the common carotid artery that forms internal and external carotid arteries. Its IA subtype indicates that the RLN will have a cephalic course and be anterior to the common carotid bifurcation, while type IB presents a cephalic course and is posterior to the common carotid bifurcation. Finally, type IC corresponds to a cephalic route that is both anterior and posterior to the bifurcation of the common carotid artery. Type II NRLN follows a route that caudally crosses the bifurcation of the common carotid artery. Subtypes IIA, IIB, and IIC are similar to subtypes IA, IB, and IC, respectively, except that they cross the bifurcation of the artery caudally. The last classification corresponds to type III, in which the recurrent laryngeal nerve is found together with the ipsilateral occurrence of NRLN. Type III subtypes are the same as those mentioned above except for the direction of the course, which can be either cephalic or caudal [13,14].

Associated with the first variation mentioned, there is a distance of 1 to 2 cm from the bifurcation of the RLN to the cricothyroid joint, but in a few patients who undergo thyroid surgery, a greater distance of 7 cm has been identified [15]. 

It should be noted that the identification of the various variations described above relies on the use of anatomical references, such as visualization of the inferior and superior thyroid arteries and the posterior suspensory ligament of the thyroid. In addition, to obtain a highly precise identification of the nerve, intraoperative neuromonitoring (IONM) is used; IONM measures nerve response using electrodes [16,17,18].

Considering the above facts, Zuckerkandl’s tubercle is a projection of the thyroid gland that is normally used as a reference point to locate the RLN. However, in extremely rare cases, in the presence of an NRLN, the tissue of the tubercle of Zuckerkandl can wrap around the RLN [19].

Therefore, it is necessary for a surgeon performing a thyroidectomy to have exhaustive knowledge of the surgical anatomy of the thyroid gland and to carefully consider the RLN and its anatomical variations; these variations are not as rare as one might think, and they significantly increase the risk of complications or injuries during surgical procedures, especially in total thyroidectomies [20,21].

In this sense, it has been shown that injuries to the RLN due to a lack of surgical precision can lead to alterations such as respiratory obstruction due to a paralysis of the vocal cords and alterations in phonation, which could be permanent. Because the inferior laryngeal nerve, which follows the RLN, innervates the muscles located in the vocal folds, an injury to either nerve portion can cause unilateral or bilateral paralysis of the vocal fold, with respiratory obstruction being the most serious consequence [21,22,23]. The unilateral affection causes dysphonia, which is characterized by the weakening of the patient’s voice and is the effect of the impediment of the union between the normal and the paralyzed folds. When the paralysis of the vocal folds is bilateral, the voice becomes weaker due to the non-movement and narrowness of the vocal folds. Likewise, the vocal folds can neither be adducted for phonation nor abducted to increase ventilation, causing stridor and respiratory obstruction. In cases of bilateral paralysis, necessary measures must be used to ensure an unobstructed airway. It is worth mentioning that the paralysis described above can occur temporarily, lasting 6 to 8 weeks, or permanently, in which case laryngeal compensation will improve voice quality [24,25].

The aim of this review was to understand the variants of the RLN and how they could influence clinical complications in the neck region.

## 2. Methodology

This systematic review and meta-analysis were carried out in Santiago, Chile, during the 2022 academic period and considered scientific articles published in English and Spanish between 1960 and 2022. For the elaboration of the structure and the contents of this review, we used as a reference the PRISMA statement for synthesis articles and review of research results. Along with this, not all the PRISMA checklist requirements are applicable due to the nature of our review. Therefore, in order to complement and contextualize the review to an anatomical format, we used the methodology proposed in the article by Henry et al., “Evidence-based anatomy methods: a guide to perform systematic reviews and meta-analyses of anatomical studies” [26]. Selected studies were screened, extracted, and evaluated using data synthesis and statistical analysis. Additionally, when available, non-narrative data were extracted and collected. Since this study did not involve any type of human material (cells, tissues, organs, patients, or others), it did not need to go through an ethics committee. This review and meta-analysis comply with the world registry of systematic reviews in the PROSPERO database with ID CRD42022357243, which can be verified at https://www.crd.york.ac.uk/prospero/#searchadvanced (Accessed 5 January 2023).

### 2.1. Search Strategy

The search was conducted in January 2022. It was based on key search terms in MEDLINE (US National Library of Medicine, National Institutes of Health) through its search interface PUBMED, WOS (it is a platform based on Web technology that collects the references of the main scientific publications of any discipline of knowledge), Google Scholar (Google Inc., Mountain View, CA, USA), CINAHL (is an index of journal articles in English and selected other languages on nursing, allied health, biomedical and medical care), Scopus (is a bibliographic database of abstracts and citations of scientific journal articles owned by Elsevier), Scielo (São Paulo State Research Support Foundation—FAPESP), and the Latin American and Caribbean Center for Information on Health Sciences. These databases were consulted from 1960 to 2022. The search strategy was designed by (JJV and MO) and validated by the Research Team to reach a final consensus and to have the keywords to be more sensitive in the search.

### 2.2. Search Terms

We used the search terms “Recurrent laryngeal nerve” (MeSH terms), “Non RLN” (no MeSH), “variation anatomical” (no MeSH), “clinical anatomy” (no MeSH), “Methodology anatomical” (no MeSH) and “clinical Neck” (no MeSH), and the Boolean connectors “AND”, “OR”, and “NOT” to combine them. The search was conducted on MEDLINE (https://pubmed.ncbi.nlm.nih.gov/?term=Recurrent+laryngeal+nerve+OR+non+recurrent+laryngeal+nerve+OR+variants+anatomical+AND+clinical+anatomy&filter=hum_ani.humans (Accessed 5 January 2023)). The search algorithm is shown in Figure 1 (Figure 1). The only search term included in the PUBMED (MeSH) thesaurus is “Nerve, laryngeal recurrent: Branches of the vagus (tenth cranial) nerve. The recurrent laryngeal nerves originate more caudally than the superior laryngeal nerves and follow different paths on the right and left sides. They carry efferents to all muscles of the larynx except the cricothyroid and carry sensory and autonomic fibers to the laryngeal, pharyngeal, tracheal, and cardiac regions”.

### 2.3. Eligibility Criteria of Included Studies

Eligibility criteria were defined using the PICOS approach, which specified the population, intervention, comparator, and outcomes relevant to the review. In this approach, the population or sample studied was individuals who underwent RLN dissections or imaging; the intervention was the identification of RLN variants, and the comparison was made between NRLN variants. Finally, the result was the clinical correlations of the variants found. This review included articles published in English in peer-reviewed journals and indexed in some of the databases mentioned in the previous section, including research articles, research reports, and original research that focused on the presence of RLN and NRLN variants and their association with clinical conditions. Review articles were excluded from the report of results, but the most relevant and similar ones were included in the discussion item. Communications to congresses were also included.

### 2.4. Outcomes

The main objective of this review was to analyze the morphology and prevalence of RLN variants and their correlation with pathologies of the anterior cervical region. This analysis was conducted descriptively in relation to normal anatomy or based on two classifications proposed in this review. Variants were defined according to the following criteria: the presence of accessory ramifications of the RLN, changes in their origin, or changes in the RLN route.

### 2.5. Data Extraction and Synthesis

The researcher (AB) imported all the retrieved studies into the Mendeley reference management software (which is a bibliographic manager that combines a web version with a desktop version). Three reviewers (MN, CC, and VB) independently screened all abstracts and completed full-text reviews of potentially relevant studies that fit the eligibility criteria. Subsequently, data were extracted from the written text, including the author and year, type of study, type of sample, prevalence, statistical values reported as results in the study, geographical region of the sample, distribution of the sample by sex, and whether the variant presented unilaterality or bilaterality. All these data were grouped for a later synthesis and reporting in the results.

### 2.6. Assessment of the Methodological Quality of the Included Studies

Data were extracted by four researchers (MN, CC, JV, and VB). For each study, the following information was extracted: last name of the first author, year of publication, type of study, sample size, prevalence, morphological characteristics of RLN, statistical data presented by the studies, geographic region, sex of the sample, and laterality. Any disagreement was resolved by consensus with a fifth investigator (AB). All articles analyzed in full-text were evaluated through quality assessment and risk of bias analysis using the methodological quality assurance tool for anatomical studies—Anatomical Quality Assurance (AQUA)—proposed by the International Evidence-Based Anatomy Working Group [27]; see Table 1 and Figure 2.

### 2.7. Analysis of Data

The data extracted in the meta-analysis were analyzed to determine the prevalence of RLN variants and their comparison with NRLN using the JAMOVI software 5.13 [64]. To analyze the relationship between RLN and NRLN, we used the REVMAN 5 software [65]. The DerSimonian–Laird model with a Freeman–Tukey double arcsine transformation was used to combine the summary data. In addition, a random effects model was used because the RLN prevalence data were highly heterogeneous. We assessed the degree of heterogeneity between included studies using the chi-squared test (χ^2^) and the heterogeneity (I²) statistic. A Q *p* value < 0.10, as proposed by The Cochrane Collaboration, was considered significant for the χ^2^ test. Values of the I² statistic were interpreted as follows with a 95% confidence interval [CI]: 0–40% indicated non-significant heterogeneity, 30–60% indicated moderate heterogeneity, 50–90% indicated substantial heterogeneity and 75–100% indicated a significant amount of heterogeneity [65].

## 3. Results

### 3.1. Included Studies

The search yielded a total of 1106 articles that met the search criteria in the title or in the abstract. After removing duplicates, 1053 articles were evaluated in full text for inclusion eligibility in the meta-analysis. Of these, 1008 were excluded because they did not meet the study’s primary and secondary outcomes or failed to meet the established review criteria. Finally, a total of 41 articles were used in the analysis (N = 29,218 patients), with imaging and cadaveric data reported descriptively in Table 2. The methodological quality and risk of bias of the included studies were evaluated using the AQUA tool and presented graphically in Figure 2.

### 3.2. Characteristics of the Studies and the Population Studied

Among the 41 studies included, 18 were case studies, 12 were retrospective studies, 7 were prospective studies, and 4 were case series. The total number of patients was 29,218, with a mean of 713 and a standard deviation (σ) of 1464.35. If we isolate the case studies and only consider the studies with an N greater than one, N was 29,200, with a mean of 1270 and a σ of 1853.47. Regarding the geographical distribution of the sample included in the studies, they were present in all continents except Africa. In Europe, 20 studies were found, which is equivalent to 48.78% of the studies included in this review. The cumulative N of patients in these 20 studies was 4009, which is equivalent to 13.72% of the total number of patients included in this review. In Asia, 16 studies were found, which is equivalent to 39.02% of the studies included in this review. It should be noted that the cumulative N of patients in these 16 studies was 25,014, which is equivalent to 85.61% of the total number of patients included in this review. In North America, two studies were found, which is equivalent to 4.88% of the studies included in this review. The cumulative N of patients in these 2 studies was 56, which is equivalent to 0.19% of the total number of patients included in this review. In South America, two studies were found, which is equivalent to 4.88% of the studies included in this review. The cumulative N of patients in these two studies was two, which is equivalent to 0.007% of the total number of patients included in this review. In Oceania, one study was identified, which is equivalent to 2.43% of the studies included in this review. The cumulative N of patients in this study was 137, which is equivalent to 0.47% of the total number of patients included in this review. These results are summarized in Figure 3 and Figure 4.

Regarding the sex of the population studied in the articles with a total sample size equal to 1, the following characteristics were identified: 5 subjects were men (27.7%), 12 subjects were women (66.66%), and 1 study did not report the sex of the sample (5.55%). In relation to the 23 studies that presented a sample size greater than 1, the following data were collected: 5 studies [8,11,13,28,29] did not specify the sex of the subjects with a total sample size of 6531 (22.36% of the total sample of studies with a sample greater than 1; 2 studies [31,32] presented only women in their sample, which is equivalent to 10 (0.034% of the total sample of studies with a sample size greater than 1); finally, 16 studies [3,11,32,33,34,35,36,37,38,39,40,41,42,43,44,45] included both men and women in their sample and the following data were obtained: the total number of men was 4849 (16.6%) and that of women was 17,810 (60.99%). Individually, in the studies that presented both men and women, the percentage of men varied between 4.08% and 81% and had a mean of 26.64, while the percentage of women varied between 19% and 95.92% with a mean of 73.36% (see Table 3).

## 4. Prevalence

Anatomical variations are anomalies in the arrangement and shape of different anatomical structures; they do not represent a pathological process or a risk to individuals since they often have a defined and functional shape. Our investigation found 23 studies that reported a prevalence greater than 1 or a prevalence with *n* > 1 of RLN variants. For analysis of data, these studies were grouped into three categories: studies with a prevalence rate greater than 10%, studies with a prevalence rate of less than 10%, and studies with *n* > 1 but with 100% prevalence. Eight studies had *n* > 1 and a prevalence of 100%; these studies [30,31,36,37,43,44,46,67] had an N ranging between 2 and 294 and a mean of 64.12 subjects. Six studies [32,34,39,40,47,66] had a prevalence greater than 10% and a deviation between 49 and 2404 subjects, and a mean of 1136.7 subjects. The prevalence in these studies ranged between 28.7% and 73.72%, with a mean of 41.36%. Finally, we found nine studies with a prevalence rate of less than 10% [3,13,16,29,33,36,38,41,42]. The prevalence in these studies ranged from 0.24% [12] to 6.32% [3], with a mean prevalence of 3.28%. Please see Table 4 for more information.

## 5. Meta-Analysis

Among the nine studies included in the forest plot meta-analysis (N = 21,867 participants), only studies that met the criteria of having a prevalence of less than 10% were grouped. This criterion correlates with the description of an anatomical variant and has no variability that, from a methodological point of view, complies with the study model that shows the prevalence of a structure in each population. The pooled prevalence of RLN variant studies in the RLN and NRLN groups was 4.61% in favor of NRLN with *p* < 0.00001 (95% CI). Heterogeneity between studies for this outcome was high (χ^2^ = 96%). The following studies did not report differences in prevalence between RLN and NRLN: [16,28,29,30,31,33,36,48,68], showing a mean prevalence of 0.35 and a CI of 0.09–1.42, 1.27 with a CI of 0.32–5.11, 0.92 with a CI of 0.31–2.80, 0.76 with a CI of 0.37–1.54, and 2.46 with a CI of 0.62–8.48, respectively. The following studies showed a high prevalence in favor of the RLN variant [44,67], with a mean of 0.12 and a CI of 0.04–0.39 and 0.06 with a CI of 0.03–0.11, respectively. Finally, the following studies showed a prevalence for the NRLN: [3,43] with a respective mean of 28.62 and a CI of 9.49–86.34 and 6.45 with a CI of 4.28–9.73.

Regarding the weight of the studies, the weight of the population, the one with the lowest was 0.7% [3,43], and the one with the highest weight was 30.1% [39] (see Figure 5). For the statistical analysis of the prevalence of the RLN variant, a forest plot was made for the 15 studies that met the condition of having a prevalence of less than 100%. The prevalence was 12% among the included studies (95% CI; SD of 0.11 to 0.14). The I² test showed an extremely high heterogeneity of studies (99.62%). The studies had a significantly homogeneous weight except for that by Gurleyik et al. (2018), which presented the highest SD (0.20 to 0.46). The CI reported by the forest plot was from 0.105 to 0.143 [34] (Figure 6).

## 6. Clinical Consideration

Regarding the clinical relationships reported by the studies included in this review, three studies did not report any type of clinical consideration related to the RLN and NRLN variants [45,49]. Thirty-eight studies reported clinical considerations associated with RLN and NRLN variants, which are as follows: 16 studies, within their results, reported possible injuries arising from surgical complications associated with surgeries such as thyroidectomies, parathyroidectomies, esophagectomies, laryngeal surgeries, or any other type of surgery of the cervical region surrounding the RLN and NRLN. These complications were reported as intraoperative lesions of the RLN and NRLN or, in some cases, alterations of the inferior thyroid artery (ITA) [8,11,14,31,32,34,35,36,38,41,42,43,50,51,52,66]. Three studies, within their clinical considerations, reported that the presence of RLN and NRLN variants could be associated with iatrogenic compromise of the vocal cords. In addition, these authors propose that when this variant is present, a medical team should perform an evaluation of the mobility of the vocal cords, especially before performing any type of laryngeal surgery [7,45,46,53,54,55,56,57]. Surgeons must be aware of considerable alterations in the anatomy of the RLN. Two studies reported that intraoperative neuromonitoring is important in re-operations associated with thyroidectomies since distortions of the anatomical planes can occur and produce secondary variants in the courses of the nerves due to the removal and/or abnormal scarring of tissues [34,66]. Furthermore, two studies reported that it is important to consider the possible variants of the RLN or the NLRN based on a timely physical and imaging examination, which could reduce iatrogenic damage such as injury to the ITA, structures that run through the tracheoesophageal groove, or by last structures associated with innervation of the RLN [11,58]. One study reported that clinical and preoperative evaluation is important because the presence of NRLN is associated in some cases with dysphagia and silhouette abnormalities visible on radiographs in the mediastinal region, thus avoiding surgical complications [59]. One study reported preoperative paralysis of the vocal cords due to the compression of a variant of the RLN route associated with a thyroid tumor [60]. Finally, one study reported, as a clinical consideration, that it should be clear that a variant of RLN could injure or alter a neighboring structure due to excessive neck movement or, many times, even within the normal ranges of movement [40].

## 7. Discussion

Knowing the anatomy of RLN is important for different health professionals, especially surgeons who work in the area where the RLN is located and some other specialists involved in the rehabilitation of the cervical region. In this study, our focus was on understanding the different anatomical variants of the RLN and how they can influence some laryngeal and neck pathologies. We also explored the prevalence reported by different studies across various population groups. We used two classifications to describe the different anatomical variants of the RLN. Based on the classification by Bula et al., 2015, the articles that met the criteria for inclusion in this classification were mostly associated with type IA and type IIA, which describe NRLNs that follow a cephalic path and surpass the union of the common carotid artery anterior to it. IIA crossed caudally and was anterior to the bifurcation of the common carotid artery. From an analysis perspective, we believe that it is important to know about this variant since the NRLN, passing anterior to the common carotid artery, could produce various iatrogenic or surgical injuries to this important blood vessel in the anterior region of the neck. Among the studies that met the criteria to be included in this classification, we found that the results were substantially heterogeneous since most of the samples were one of the types; however, we can highlight that the type that presented the highest prevalence was type II in which two branches of the RLN are observed separately up to the point of entry into the tracheoesophageal groove, which shows that the arrangement of branches could be variable and could have associated surgical complications. 

In this review, we analyzed 18 case studies; and although we know that the RLN variants in this review were found by chance and not from pure investigative intention, we believe that the use of these studies could provide us with information on some type of special variant or one that is different from those proposed theoretically or by the classifications included in this study. In relation to the above, a case study was included that described the presence of an aberrant right subclavian artery (ARSA), an NRLN, and a type I right thoracic duct draining into the right brachiocephalic vein, which is not commonly found within the classical literature or published in a scientific article [67] (see Figure 7). 

Regarding the geographical distribution of the sample, the largest number of studies included in this review were from the European continent with a cumulative N = 4009 subjects. However, it should be noted that although most of the studies were conducted in Europe, the sample with the highest number of N was found in the Asian continent (N = 25,014 subjects). Although the sample number is quite high in the two continents, these data alone and the data proposed by the studies do not allow us to associate the NRLN and RLN variant with any specific race or ethnicity. In addition, within the results or discussion of the articles included, none of them carried out an analysis on or attributed the variant to any specific type of race or continent. Although the study by Makay et al., 2008, which was carried out on the European continent, presented a high prevalence, this prevalence could not be inferred or classified as a condition for the aforementioned. 

Sex was declared in most of the studies, and this declaration reveals some important characteristics that we can mention. Women had a 3:1 predominance over men in the case studies, which could be an indication that the RLN or NRLN variant is more common in women than men. However, for the studies with N > 1, in 25% of the accumulated sample, the authors did not specify the sex of their samples. This was because some studies had only included segments of the neck region in their population samples and had not identified the sex or age of individuals. In the studies that did report the sex of their patients, we identified a possible bias which is related to the fact that in some studies, only the sex of the sample that presented the variant was reported and not the sex of the entire sample. Although this reporting can be interpreted on a case-by-case basis, this relationship does not provide conclusive data to determine whether there is a variant that is more prevalent in men or women. 

Regarding the numerical results of 16 studies that reported their entire sample and divided them into men and women, a 3:1 ratio was presented in favor of a greater number of women, with a cumulative N = 22,659 neck regions studied. In these studies, it could be argued that there is a relationship between the variant and the female sex, which could be attributed to the higher incidence of thyroid cancer surgeries or diagnostic tests performed in women than men in the region [69]. Due to this relationship, there could be a greater probability of finding NRLN or RLN variants during surgery. However, we believe that in order to make a categorical statement about this relationship, additional studies are needed that focus on determining the prevalence of this variant by sex. 

Of the studies included in this review, to analyze their methodological quality, they were subjected to the Quality Assessment of Anatomical Studies bias analysis [27]. As a result of the five domains covered by this tool, it was reported that the criteria of objectives and characteristics of the studies and the criteria of anatomical description had a low risk of bias and that for the design criteria, methodological characteristics, and result reporting risk of bias was high. The foregoing shows that three criteria out of five presented a high risk of bias, which a priori tells us that the results reported by several of the studies should be taken with caution. We must emphasize that in some domains, the level of risk of bias is increased by case studies that sometimes do not have a rigorous methodological quality. 

To avoid bias, we excluded all studies that had an N equal to one [61] and studies reporting a prevalence of 100% [54,62,63] because the sample was chosen intentionally or for convenience and therefore gave this result. Fifteen studies met the criteria to be meta-analyzed based on the prevalence variable. However, they presented high heterogeneity, which indicates that they were highly disparate in their sample; nevertheless, they also met the criteria defined by the research group for meta-analysis. The results obtained in the meta-analysis showed that the prevalence of the RLN and NRLN variants was 0.12 with a CI of 0.11 to 0.14. From the theoretical point of view of the anatomical variant definition, this prevalence would leave it out and represent it as a variability of RLN or NRLN. We believe that this is because six of the meta-analyzed studies presented a prevalence above 20%, which is represented as our standard error, which was 0.0097. Therefore, the reported prevalence data must be analyzed individually and with caution because any other factor may have intentionally or accidentally influenced prevalence in each of the included studies. The meta-analysis showed that the prevalence results were statistically significant in relation to the comparison between the different samples (*p* = 0.001). For the meta-analysis, the weight of the included studies fluctuated between 1.66% and 7.96%, which correlates with the high heterogeneity of the studies included for the meta-analysis, supported by the differences in total N included. 

For the forest plot, the prevalence of RLN was compared to that of NRLN, and the data were presented as prevalence or the number of variants reported by each of the studies in relation to the total N of the sample. The foregoing was applied to all the studies that met the condition of declaring the RLN and NRLN variant in their sample jointly. Applying the aforementioned criteria, the variant in the RLN branches was taken as a control group and the NRLN as a comparison group, with the main result that there was a high prevalence in the NRLN studies with 1.65 and a CI of 1.33 to 2.04. Although the heterogeneity reported by the forest plot was high, the results showed us that in studies that showed both variants, the NRLN was more prevalent.

In most of the included studies, the clinical considerations refer to intra-surgical complications of the anterior region of the neck, thyroidectomies, parathyroidectomies, and esophageal surgeries, among others. The foregoing is because not knowing the presence of a variant or not having found the variant in a previous diagnostic test could cause involuntary iatrogenic injuries that could produce a nerve injury or some type of vascular injury to a surrounding blood vessel. 

Another relevant clinical consideration that we believe should be carefully studied in great detail to see if the relationship is direct or accidental is that of the alteration in the vocal cords, which could be a direct clinical condition and not just an effect of a particular procedure. One of the most important clinical considerations of the RLN variant is intra-surgical complications that are especially associated with the most common surgeries, namely thyroidectomy and parathyroidectomy, in the neck area. These surgeries could impact the RLN. These surgeries are preferred for the treatment of thyroid and parathyroid cancers. Patients with these cancers have a large volume increase that can be associated with goiter of the thyroid gland and hypervascularization in the area forming collateral branches of the thyroid arteries. A variant of the laryngeal nerve can cause a redistribution in the course of the thyroid arteries or some compression of the same in the lower or upper thyroid region (Figure 8). A consideration of these variants is crucial before neck and thyroid surgery: not considering the presence of these variants could cause intraoperative bleeding. 

Lastly, one study linked NRLN to dysphagia, but the study did not explain its mechanism. Therefore, we believe that its finding was fortuitous and did not correlate with NRLN variants. Our review exhaustively examined NRLN and RLN variants, taking as a sample study that included cadavers, imaging, and individual intraoperative findings of patients with these variants. We tried not to leave out previously established studies that met our criteria. 

We believe that our review is different from the following reviews found in the literature: Bakaline et al., 2018 [68] reported about the RLN variant descriptively and compared it to other vascular variants such as ITA and ARSA. In contrast, our study was mainly focused on the RLN and the NRLN variants and reported their individual and comparative prevalence and how these variants influence a clinic. We thus employed a translational approach. Henry et al., 2016, carried out a comprehensive meta-analysis of the variants of the extra-laryngeal branches of the RLN. Although we carried out an exhaustive analysis of variants in the branches of the RLN, our review made a comparison between different edges in great depth, as mentioned above. Ling and Smoll, 2016 [70], carried out a systematic review based only on RLN classifications; they did not evaluate methodological quality or make clinical comparisons, which means that our study approached this variant more comprehensively and in more descriptive and quantitative dimensions than Ling and Smoll. Lastly, Polednak et al., 2017 [71] and Polednak et al., 2019 [72], carried out a descriptive analysis of the ITA and the NRLN, respectively, but provided no further descriptions. Taking into consideration the differences listed above, we hope that our review and meta-analysis will provide a broad analysis of the NRLN and RLN variants and offer a resource for medical or rehabilitation professionals of the neck region.

## 8. Limitations

The limitations of this review are the publication bias of the included studies since studies with different results that were in non-indexed literature in the selected databases may have been left out, the probability of not having carried out the most sensitive and specific search regarding the topic to be studied, and finally, personal sessions of the authors for the selection of articles.

## 9. Conclusions

The authors of this study believe that this meta-analysis provides an updated understanding of the prevalence of RLN variants and highlights some clinical correlations, such as intra-surgical complications and some pathologies and functional aspects of the vocal cords, which could offer a guideline in management prior to surgery or be of interest in the diagnosis of pathologies of the vocal cords. However, we caution that these results should be taken with care due to the high bias of some included studies. Finally, we believe that it is necessary to carry out new studies to investigate the functional implications of these variants and to furnish a high methodological quality to establish an anatomy–clinical relationship between the NRLN or the RLN with pathologies of the neck region. The new studies will enable us to validate these findings more effectively. 

## Figures and Tables

**Figure 1 life-13-01077-f001:**
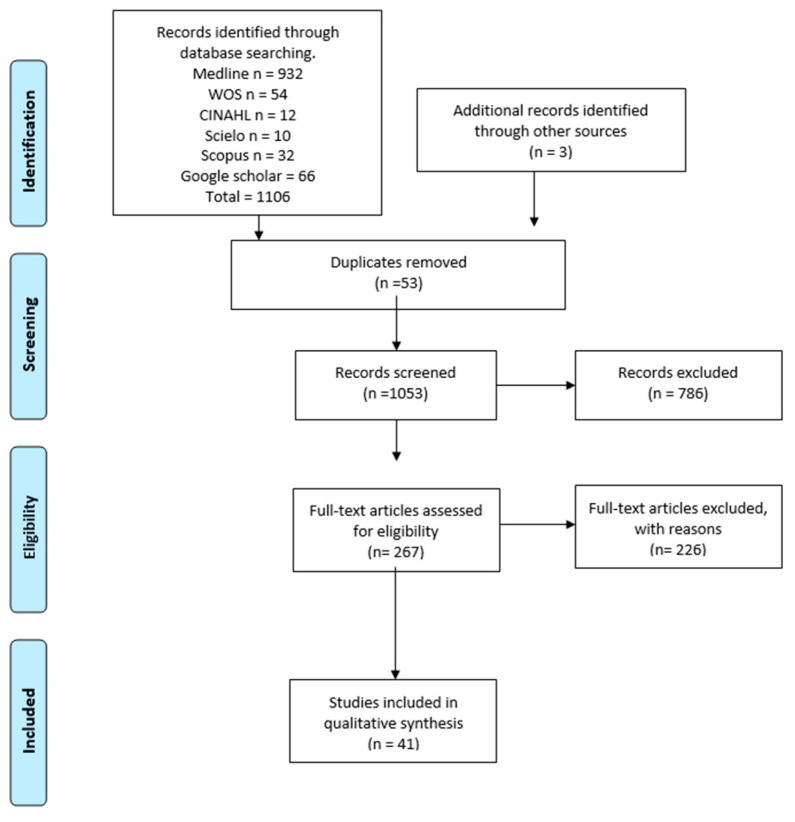
Search diagram.

**Figure 2 life-13-01077-f002:**
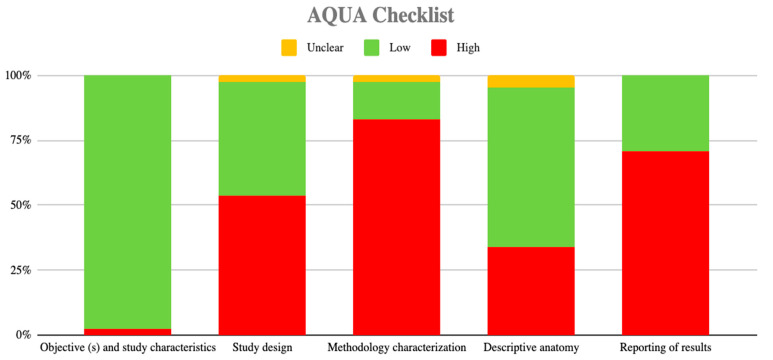
Risk of bias graph.

**Figure 3 life-13-01077-f003:**
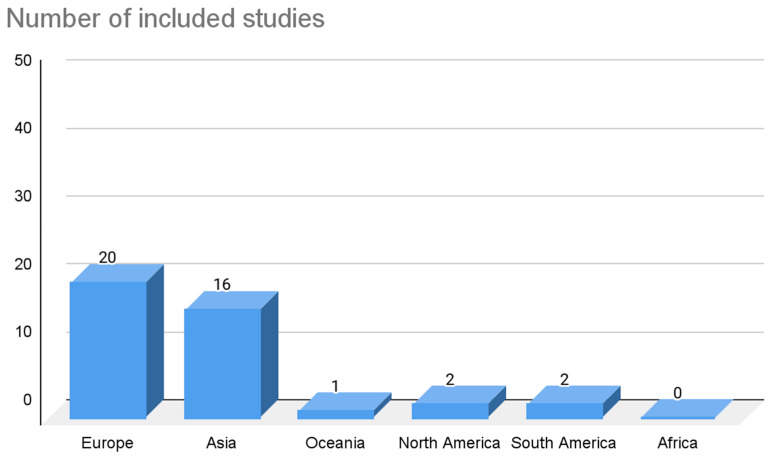
Graph of the number of included articles distributed by geographic region.

**Figure 4 life-13-01077-f004:**
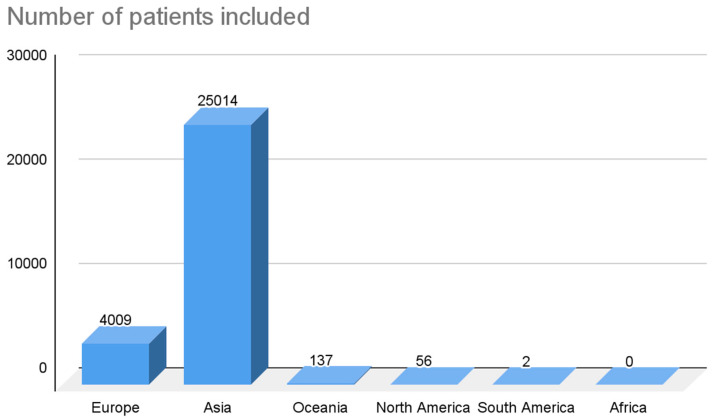
Graph of the number of patients in the included articles distributed by geographic region.

**Figure 5 life-13-01077-f005:**
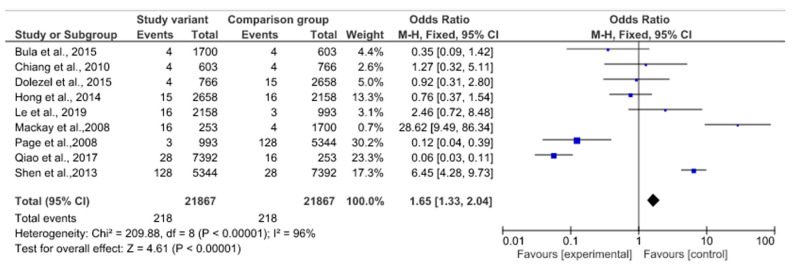
Forest plot of the relationship between the NLRN and RLN [3,12,15,29,33,35,38,41,42].

**Figure 6 life-13-01077-f006:**
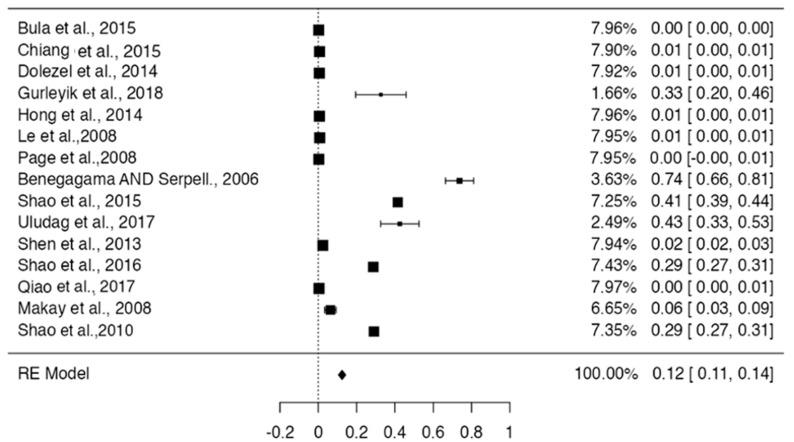
Forest plot of prevalence of RLN variations of the included studies [3,7,12,15,29,32,33,35,38,39,40,41,42,61,63].

**Figure 7 life-13-01077-f007:**
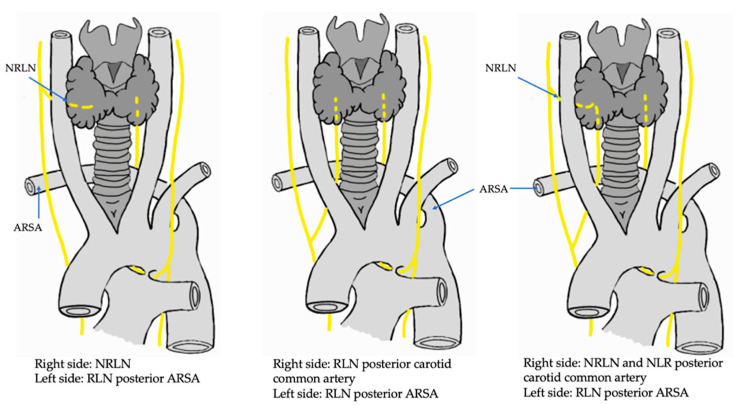
Drawing of the variants RLN and NRLN, in association with ARSA. RLN: recurrent laryngeal nerve; NRLN: non-recurrent laryngeal nerve; ARSA: aberrant right subclavian artery.

**Figure 8 life-13-01077-f008:**
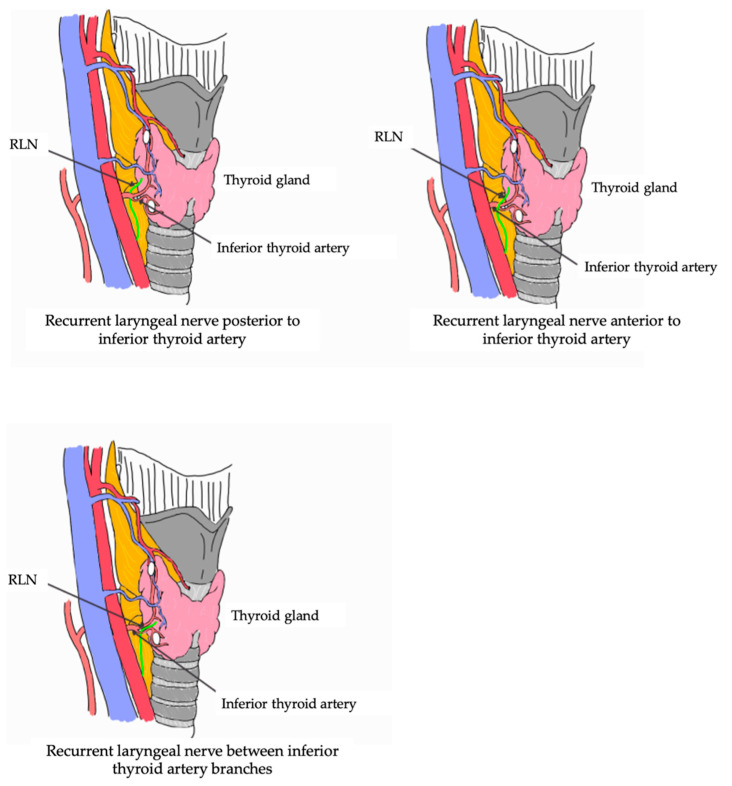
Trajectory RLN and its relationship with the vascularization of the thyroid region. RLN: recurrent laryngeal nerve.

**Table 1 life-13-01077-t001:** Anatomical quality assurance checklist.

Authors	Study Design	Domain 1				Domain 2					Domain 3						Domain 4					Domain 5				
		1	2	3	4	5	6	7	8	9	10	11	12	13	14	15	16	17	18	19	20	21	22	23	24	25
[3]	Prospective study	Y	Y	Y	Y	Y	N	Y	Y	Y	Y	N	Y	N	Y	Y	Y	Y	N	Y	Y	Y	Y	N	NA	Y
[13]	Retrospective study	Y	Y	Y	Y	Y	Y	Y	N	Y	Y	Y	Y	N	N	Y	Y	Y	Y	Y	N	Y	Y	Y	NA	Y
[14]	Case report	Y	Y	N	Y	N	Y	N	Y	Y	N	Y	Y	N	Y	Y	Y	Y	Y	Y	N	Y	Y	Y	NA	N
[28]	Retrospective study	Y	Y	Y	Y	Y	N	Y	Y	Y	Y	Y	Y	N	Y	Y	Y	N	Y	Y	Y	Y	Y	Y	NA	Y
[29]	Multiple-case study	Y	Y	Y	Y	N	Y	Y	N	N	Y	N	N	Y	N	N	Y	Y	N	Y	Y	N	Y	N	NA	Y
[30]	Prospective study	Y	N	Y	Y	N	Y	Y	Y	Y	N	Y	Y	N	Y	Y	Y	Y	Y	Y	N	Y	Y	Y	NA	Y
[31]	Multiple-case study	Y	Y	Y	Y	N	Y	Y	N	Y	Y	N	N	Y	Y	Y	Y	Y	N	Y	Y	N	Y	N	NA	Y
[32]	Prospective study	Y	Y	Y	N	Y	Y	N	N	Y	Y	N	Y	N	N	Y	U	Y	Y	U	Y	Y	Y	Y	NA	NY
[33]	Retrospective study	Y	Y	Y	Y	Y	N	Y	Y	Y	Y	N	Y	N	Y	Y	Y	Y	N	Y	Y	Y	Y	N	NA	Y
[34]	Retrospective study	Y	Y	N	Y	Y	Y	Y	Y	N	Y	Y	Y	Y	Y	Y	Y	N	Y	Y	N	Y	Y	N	NA	Y
[35]	Retrospective study	Y	Y	Y	Y	Y	Y	Y	N	Y	Y	Y	U	U	Y	N	Y	Y	Y	N	Y	Y	N	Y	NA	Y
[36]	Retrospective study	Y	Y	Y	Y	N	Y	N	Y	Y	Y	Y	Y	N	Y	Y	Y	Y	Y	Y	N	Y	Y	Y	NA	Y
[37]	Multiple-case study	Y	Y	Y	Y	N	Y	Y	N	N	Y	Y	Y	Y	N	Y	Y	Y	N	Y	Y	Y	Y	N	NA	Y
[38]	Retrospective study	Y	Y	N	Y	Y	Y	Y	Y	N	Y	Y	Y	Y	Y	Y	Y	N	Y	Y	N	Y	Y	N	NA	Y
[39]	Retrospective study	Y	Y	Y	Y	Y	N	Y	Y	Y	Y	N	Y	N	Y	Y	Y	Y	N	Y	Y	Y	Y	N	NA	Y
[40]	Retrospective study	Y	Y	Y	Y	Y	N	Y	Y	Y	Y	N	Y	N	Y	Y	Y	Y	N	Y	Y	Y	Y	N	NA	Y
[41]	Retrospective study	Y	Y	Y	Y	Y	N	Y	Y	Y	Y	N	Y	N	Y	Y	Y	Y	N	Y	Y	Y	Y	N	NA	Y
[42]	Retrospective study	Y	Y	Y	Y	Y	N	Y	Y	Y	Y	N	Y	N	Y	Y	Y	Y	N	Y	Y	Y	Y	N	NA	Y
[43]	Retrospective study	Y	Y	Y	Y	Y	N	Y	Y	Y	Y	N	Y	N	Y	Y	Y	Y	N	Y	Y	Y	Y	N	NA	Y
[44]	Prospective clinical trial	Y	Y	Y	Y	Y	N	Y	Y	Y	Y	N	Y	N	Y	Y	Y	Y	N	Y	Y	Y	Y	N	NA	Y
[45]	Case report	Y	Y	Y	Y	N	Y	N	Y	Y	N	Y	Y	N	Y	Y	N	Y	N	Y	N	Y	Y	Y	NA	N
[46]	Prospective study	Y	Y	Y	Y	Y	Y	N	N	Y	Y	Y	N	Y	Y	Y	Y	N	N	Y	Y	Y	Y	Y	NA	Y
[47]	Multiple-case study	Y	Y	Y	Y	N	Y	Y	N	N	Y	N	N	Y	N	N	Y	Y	N	Y	Y	N	Y	N	NA	Y
[48]	Case report	Y	N	Y	Y	N	Y	Y	Y	Y	N	Y	N	N	N	Y	Y	N	N	Y	N	Y	N	Y	NA	Y
[48]	Case report	Y	Y	Y	Y	N	Y	N	Y	Y	N	Y	Y	N	Y	Y	N	Y	N	Y	N	Y	Y	Y	NA	N
[49]	Case report	Y	Y	Y	Y	N	Y	N	Y	Y	N	Y	Y	N	Y	Y	N	Y	N	Y	N	Y	Y	Y	NA	Y
[49]	Prospective study	Y	Y	Y	Y	Y	N	Y	Y	Y	Y	N	Y	N	Y	Y	Y	Y	N	Y	Y	Y	Y	N	NA	Y
[50]	Case report	Y	N	Y	Y	N	Y	N	Y	Y	N	Y	Y	N	Y	Y	N	Y	N	Y	N	Y	Y	Y	NA	N
[51]	Case report	Y	Y	Y	Y	N	Y	N	Y	Y	N	Y	Y	N	Y	Y	N	Y	N	Y	N	Y	Y	Y	NA	N
[52]	Case report	Y	Y	Y	Y	N	Y	N	Y	Y	N	Y	Y	Y	Y	Y	N	Y	N	Y	N	Y	Y	Y	NA	Y
[53]	Case report	Y	Y	Y	Y	N	Y	N	Y	Y	N	Y	Y	N	Y	Y	Y	Y	N	Y	Y	Y	Y	Y	NA	Y
[54]	Case report	Y	Y	Y	Y	N	Y	N	Y	Y	N	Y	Y	N	Y	Y	Y	Y	N	Y	Y	Y	Y	Y	NA	Y
[55]	Case report	Y	Y	Y	Y	N	Y	Y	Y	Y	N	Y	Y	Y	Y	Y	Y	Y	Y	N	N	N	Y	Y	NA	N
[56]	Case report	Y	Y	Y	Y	N	Y	N	Y	Y	Y	N	Y	N	N	Y	Y	N	N	N	Y	Y	N	N	NA	Y
[7]	Prospective study	Y	Y	Y	Y	Y	Y	Y	Y	N	Y	Y	N	Y	Y	N	Y	N	Y	Y	N	Y	N	Y	NA	Y
[57]	Case report	Y	Y	Y	Y	Y	N	Y	Y	Y	N	N	Y	Y	Y	Y	Y	Y	Y	Y	Y	N	Y	Y	NA	N
[58]	Case report	Y	Y	Y	N	Y	Y	N	N	Y	N	N	Y	N	Y	N	Y	N	N	N	Y	N	N	Y	NA	Y
[59]	Case report	Y	Y	Y	Y	Y	N	N	Y	Y	Y	N	N	N	Y	Y	U	Y	Y	Y	U	Y	N	Y	NA	Y
[60]	Case report	Y	N	N	Y	N	Y	N	Y	Y	N	Y	Y	N	Y	Y	N	Y	N	Y	N	Y	Y	Y	NA	N
[61]	Case report	Y	Y	Y	N	Y	Y	N	N	Y	Y	N	Y	N	N	Y	N	Y	Y	N	Y	N	Y	N	NA	N
[62]	Case report	Y	Y	Y	N	Y	Y	U	U	N	Y	Y	N	Y	N	N	Y	Y	Y	N	Y	Y	N	N	NA	N
[63]	Prospective study	Y	Y	Y	Y	Y	Y	Y	Y	Y	Y	N	Y	N	Y	Y	Y	Y	N	Y	Y	Y	Y	N	NA	Y

Domains and Questions: Domain 1: objective(s) and subject characteristics. (1) Was (Were) the objective(s) of the study clearly defined? (2) Was (Were) the chosen subject sample(s) and size appropriate for the objective(s) of the study? (3) Are the baseline and demographic characteristics of the subjects (age, sex, ethnicity, healthy or diseased, etc.) appropriate and clearly defined? (4) Could the method of subject selection have in any way introduced bias into the study? Domain 2: study design. (5) Does the study design appropriately address the research question(s)? (6) Were the materials used in the study appropriate for the given objective(s) of the study? (7) Were the methods used in the study appropriate for the given objective(s) of the study? (8) Was the study design, including methods/techniques applied in the study, widely accepted or standard in the literature? If “no”, are the novel features of the study design clearly described? (9) Could the study design have in any way introduced bias into the study? Domain 3: methodology characterization. (10) Are the methods/techniques applied in the study described in enough detail for them to be reproduced? (11) Was the specialty and the experience of the individual(s) performing each part of the study (such as cadaveric dissection or image assessment) clearly stated? (12) Are all the materials and methods used in the study clearly described, including details of manufacturers, suppliers, etc.? (13) Were appropriate measures taken to reduce inter- and intra-observer variability? (14) Do the images presented in the study indicate an accurate reflection of the methods/techniques (imaging, cadaveric, intraoperative, etc.) applied in the study? (15) Could the characterization of methods have in any way introduced bias into the study? Domain 4: descriptive anatomy. (16) Were the anatomical definition(s) (normal anatomy, variations, classifications, etc.) clearly and accurately described? (17) Were the outcomes and parameters assessed in the study (variation, length, diameter, etc.) appropriate and clearly defined? (18) Were the figures (images, illustrations, diagrams, etc.) presented in the study clear and understandable? (19) Were any ambiguous anatomical observations (i.e., those likely to be classified as “others”) clearly described/depicted? (20) Could the description of anatomy have in any way introduced bias into the study? Domain 5: reporting of results. (21) Was the statistical analysis appropriate? (22) Are the reported results as presented in the study clear and comprehensible, and are the reported values consistent throughout the manuscript? (23) Do the reported numbers or results always correspond to the number of subjects in the study? If not, do the authors clearly explain the reason(s) for subject exclusion? (24) Are all potential confounders reported in the study, and subsequently measured and evaluated, if appropriate? (25) Could the reporting of results have in any way introduced bias into the study? (Tomaszewski et al. [13]).

**Table 2 life-13-01077-t002:** Features of the included articles.

Author & Year	Type of Study & N	Incidence and Characteristics	Statistical Values	Geographic Region	Laterality	Gender
[12]	Retrospective study	4 patients with NRLN, 1400 patients with RLN bilateral exposition, 300 witch unilateral exposition of RLN	Not presented	Poland	Uni & bilateral	Not mentioned
[13]	Case report (1 patient)	1 case of right NRLN type I	Not presented	Turkey	Right	1 woman (100%)
[3]	Nonrandomized prospective study (253 patients)	57 RLN with 2 branches 4 RLN with 3 branches Position of RLN respect to inferior thyroid artery (ITA) Posterior: Right: 163 Left: 181 Anterior: Right: 61 Left: 50 Between the inferior thyroid artery branches Right: 19 Left: 13	The distributions of the three types of relationships between the RLNs and the ITA differ significantly (*p* < 0.05). There was no statistically significant difference between the sides regarding the relationship of RLNs with ITA (*p* > 0.05).	Turkey	bilateral	205 men (81%) & 48 women (19%)
[28]	Retrospective study	603 patients, 1 NRLN type I and 3 NRLN type II	Not presented	Korea	Right	Not mentioned
[29]	Case series (4 patients)	4 NRLN (1 type II A)	Not presented	Vietnam	Right	4 women (100%)
[30]	Prospective study (2158 patients)	16 patients with NRLN	Not presented	Vietnam	Not mentioned	14 women (88%) & 2 men (12%)
[31]	Case series (6 bodies)	Left RLN average length was 136.6 mm Right RLN average length was 75 mm	A value *p* < 0.05 was considered statistically significant	France	Bilateral	6 women (100%)
[32]	Prospective study (137 patients)	1 right NRLN. RLN bifurcation before entering the larynx on the right side, 49 cases (35.77%) RLN bifurcation before entering the larynx on the left side, 28 cases (20.32%) Ramifications on both sides of the RLN, 14 cases (10.22%)	There was a statistically significant difference between the variations due to bifurcation on the right side and the bifurcation on the left side (*p* ≤ 0.05).	Australia	Bilateral	114 women (83.21%) & 23 Men (16.79%)
[33]	Retrospective study	766 patients, 1439 RLN (725 right and 714 left) and 4 NRLN type IIA	Not presented	France	Uni & bilateral	615 women (80.2%) & 151 men (19.8%)
[34]	Retrospective study	49 patients with 61 RLN (38 right and 23 left)	Not presented	Turkey	Uni & bilateral	47 females (96%) & 2 men (4%)
[35]	Retrospective study (2658 patients)	15 patients with NRLN	Not presented	Korea	Not mentioned	2179 women (82%) & 479 men (18%)
[36]	Retrospective study (16 patients)	16 cases of NRLN	Not presented	Korea	Right	14 women (88%) & 2 men (12%)
[37]	Case series (100 bodies)	11 left RLN separates above the aortic arch crossing it with a considerable distance from the vagus nerve 48 left RLN separates at the aortic arch level 41 left RLN emerges from the vagus nerve in perpendicular direction under the aortic arch	There was no association between sex and course of RLN (*p* = 0.386).	Austria	Left	72 men (72%) & 28 women (28%)
[38]	Retrospective study (993 patients)	1 patient with right NRLN type I 1 patient with right NRLN type II	Not presented	France	Right	773 women (77.84%) & 220 men (22.16%)
[40]	Retrospective study (2068 pacientes)	Variation in the site of branching to the larynx by a fan-shaped recurrent laryngeal nerve on the right side (155 cases) and left side (46 cases). Variation in the site of branching to the larynx by a recurrent laryngeal nerve with extralaryngeal branches of the two-branch subtype on the right side (129 cases) and left side (181 cases) Variation in the site of branching to the larynx by a recurrent laryngeal nerve with extralaryngeal branches of the two-branch subtype dividing the larynx separately on the right side (111 cases) and 170 cases on the left side.	There was a statistically significant difference when comparing the branching of the recurrent laryngeal nerve with a fan shape on the left and right sides (*p* < 0.001) There was a statistically significant difference when comparing the branching of the recurrent laryngeal nerve with extralaryngeal branches on the left and right sides (*p* < 0.001).	China	Bilateral	Not mentioned
[41]	Retrospective study (7392 patients)	28 Right NRLN 5 type I 19 type IIa 4 type IIb	For the enumeration data they applied the I2 test and for the measurement data they applied the 2-independent-sample *t*-test. The difference was statistically significant with *p* < 0.05. The appearance of NRNL was not significantly associated with sex, age, nature of the tumor, number of lesions, or monitor use (*p* > 0.05).	China	Right	1492 men (20.1%) & 5900 women (79.8%)
[41]	Retrospective (2068 patients)	322 RLN branched extralaryngeally According to number of branches per side: 131 extralaryngeal branching right RLNs (129 RLNs with two branches, 2 RLNs with three branches, 0 RLNs with four branches) 191 extralaryngeal branching left RLNs (181 RLNs with two branches, 7 RLNs with three branches, 3 RLNs with four branches)	Value was significantly lower than that of the variation that occurred in the left side (*p* < 0.001). The variation occurred more frequently on the right side than on the left (*p* < 0.0001)	China	Bilateral	1894 females (91.59%) y 174 males (8.41%)
[42]	Retrospective study (5344 patients)	Anatomical variations of the RLN in the cases of the dissection group (23.4%/*n* = 128) RLN with outside position (*n* = 43) RLN with anterior position (*n* = 56) RLN with two branches before entering the larynx (*n* = 29)	There were no significant differences in sex, age, thyroid disease, type of surgery, and number of surgeries between the two groups using chi-square analysis (*p* > 0.05).	China	Bilateral	1418 males (26.53%) y 3926 females (73.47%)
[43]	Retrospective study (55 cadavers)	Right side: Right RLN presents 2–5 extralaryngeal branches (89.1%). 22 bifurcated RLN, 15 trifurcated RLN, 12 multi-branched RLN, 6 unbranched RLN. Left side: left RLN presents 2–5 extralaryngeal branches (74.6%). 15 bifurcated RLN, 19 trifurcated RLN, 7 multi-branched RLN, 14 unbranched RLN.	Data were quantitatively analyzed with McNemar’s test and Fisher’s exact test. Statistical significance was determined based on a value of *p* < 0.05.	USA	Bilateral	28 males (50.9%) y 27 females (49.1%)
[44]	Prospective clinical essay (294 patients)	RLN by number of bifurcations per side: Left side: 159 with one bifurcation (type I). 60 with two bifurcations (type II). 15 with three bifurcations (type III). Right side: 170 with one bifurcation (type I). 63 with two bifurcations (type II). 11 with three bifurcations (type III). 2 right NRLN (type IV)	There was no statistically significant difference for sideways comparison (*p* > 0.05)	China	Uni & bilateral	49 men (16.7%) & 245 women (83.3%)
[45]	Case report (1 patient)	1 left NLR under Inferior Thyroid artery	Not presented	China	Left	Woman (100%)
[46]	Prospective study	67 RLN in 36 patients.	Not presented	Turkey	Uni & bilateral	30 Women (83.33%), 6 men (16.67%)
[66]	Case report	1 patient with RLN with an extralaryngeal bifurcation	Not presented	Turkey	Right	Not mentioned
[47]	Case series (2 patients)	NRLN originating from the vagus nerve, presenting 2 branches before passing through Berry’s ligament, 1 case (50%) NRLN originating in the vagus nerve, following its normal course below the trunk of the inferior thyroid artery, 1 case (50%)	Not presented	Turkey	Unilateral	Not mentioned
[48]	Case study (1 patient)	Coexistence of right RLN with NRLN type II	Not presented	Turkey	Right	1 Woman (100%)
[49]	Case study (1 body)	Coexistence of NRLN type II, aberrant subclavian artery & thoracic duct type I on the right side	Not presented	South korea	Right	1 Man (100%)
[50]	Case report (1 patient)	Left and right RLN	Not presented	Iran	Bilateral	1 woman (100%)
[51]	Case study (1 patient)	Coexistence of right RLN with NRLN	Not presented	Japan	Right	1 Man (100%)
[52]	Case study (1 patient)	Distance of 7 cm between the extralaryngeal bifurcation to the cricothyroid joint, 1 case (100%)	Not presented	Greece	Left	1 woman (100%)
[53]	Case study (1 patient)	1 patient with right NRLN	Not presented	USA	Right	1 woman (100%)
[54]	Case report	1 patient with NRLN	Not presented	Greece	Right	Woman (100%)
[55]	Case report (1 patient)	Right NRLN emerges from vagus nerve	Not presented	Rumania	Right	1 Woman (100%)
[56]	Case report (1 patient)	Left double RLN	Not presented	Turkey	Left	1 Woman (100%)
[7]	Prospective study (2404 patients)	According to extralaryngeal divergence (type I): - Left (*n* = 205) - Right (*n* = 171) According to fan-shaped divergence (type II): - Left (*n* = 61) - Right (*n* = 173) Non-recurrent laryngeal nerve (Type V): - Left (*n* = 0) - Right (*n* = 15)	The laterality of the NLR variations was not statistically significant (*p* = 0.07)	China	Bilateral	510 men (21.2%) & 1894 women (78.8%)
[57]	Case report (1 patient)	Bilateral RLN	Not presented	Japan	Bilateral	Man (100%)
[58]	Case study (1 patient)	Right NRLN with thyroid tissue in zuckerkandl tubercle region	Not presented	Brazil	Right	1 woman (100%)
[59]	Case report (1 case)	1 (100%) left NRLN	Not presented	Brazil	Left	Woman (100%)
[60]	Case report (1 patient)	Right RLN lies over the thyroid tumor and compressed against the clavicle	Not presented	Taiwán	Right	Man (100%)
[61]	Case report	1 Patient with right NRLN	Not presented	Turkey	Right	Woman (100%)
[62]	Case report	1 (100%) of NRLN	Not presented	Portugal	Right	Man (100%)
[63]	Prospective study (94 pacientes)	77 RLNs bifurcate before entering the larynx. 36 right RLN branched. 41 branched left RLN Bilateral examination (*n* = 67): 27 RLN unilaterally branched 19 bilaterally branched RLNs 21 RLN unbranched	There was no significant difference in branching rates between sides (*p* = 0.196). There was no significant difference between the sides in terms of branching (*p* = 0.471).	Turkey	Bilateral and unilateral	76 females (80.9%) y 18 males (19.2%)

**Table 3 life-13-01077-t003:** Sex distribution of the included studies.

Study	N Total	Men and %	Women and %	Does Not Specify and %
[3]	253	205/81%	48/19%	0/0%
[5]	5344	1418/26.53%	3926/73.47%	0/0%
[8]	2068	174/8.41%	1894/91.59%	0/0%
[13]	1700	0/0%	0/0%	1700/100%
[28]	603	0/0%	0/0%	603/100%
[29]	4	0/0%	4/100%	0/0%
[30]	2158	0/0%	0/0%	2158/100%
[31]	6	0/0%	6/100%	0/0%
[32]	137	23/16.79%	114/53.21%	0/0%
[33]	766	151/19.8%	615/80.2%	0/0%
[34]	49	2/4.08%	47/95.92	0/0%
[35]	2658	479/18.02%	2179/81.98%	0/0%
[36]	16	2/12.5%	14/87.5%	0/0%
[36]	100	72/72%	28/28%	0/0%
[38]	993	220/22.16%	763/77.84%	0/0%
[39]	2068	0/0%	0/0%	2068/100%
[40]	2404	510/21.2%	1894/78.8%	0/0%
[41]	7392	1492/20.2%	5900/79.8%	0/0%
[43]	55	28/50.9%	27/49.1%	0/0%
[44]	294	49/16.7%	245/53.3%	0/0%
[46]	36	6/16.77%	30/83.33%	0/0%
[47]	2	0/0%	0/0%	2/100%
[59]	94	18/19.2%	76/80.9%	0/0%
Mean Famele and Male	-	27.88%	72.12%	-

**Table 4 life-13-01077-t004:** Total subject number and RLN variations prevalence of the included studies.

Author	N Patients	Prevalence Number and Percentage
[3]	253	16/6.32%
[8]	2068	601/29.06%
[12]	1700	4/0.24%
[15]	603	4/0.66%
[27]	2658	15/0.56%
[64]	16	16/100%
[28]	4	4/100%
[11]	2158	16/0.74%
[30]	6	6/100%
[31]	137	101/73.72%
[32]	766	4/0.52%
[33]	49	16/32.65%
[36]	100	100/100%
[37]	993	3/0.30%
[38]	2068	858/41.49%
[40]	7392	28/0.38%
[41]	5344	128/2.39%
[42]	55	55/100%
[43]	294	294/100%
[45]	36	36/100%
[66]	2	2/100%
[68]	2404	690/28.7%
[58]	94	40/42.55%

## Data Availability

All the data is in the document.

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
