# Peer review of "Systematic Review and Meta-Analysis: Recurrent Laryngeal Nerve Variants and Their Implication in Surgery and Neck Pathologies, Using the Anatomical Quality Assurance (AQUA) Checklist"

_life, 2023, doi:10.3390/life13051077_

Round 1

Reviewer 1 Report

General comment

The paper is a systematic review and meta-analysis of the recurrent laryngeal nerve variants and its implication in surgery and neck pathologies. The majority of the paper is devoted to the description of the methods and not to the topic – laryngeal nerve variants. If I read the paper as a clinician there is not enough data that I can use in my everyday work. The editor should decide whether the content is interesting enough for the readers of the journal in order not to reject it. Otherwise I suggest major revision.

Detailed comments

Abstract

There are abbreviations without explanations in the abstract.

The abstract is too long. It should give just the essential information.

»Results. The included studies that 38 showed variants of the recurrent laryngeal nerve included in this review were 41, with a cumulative N of 29218.« What represents 29218? No of cases included?

Methodology

Figure 1: I have an impression that the authors have forgotten to put a total number of records in the diagram.

Table 1. Is it really necessary to show the answers on all domains for all included studies? I think it would be enough just to tell that the studies had to fulfill such and such criteria to be included in the analysis. I would expect that the numbers of the references in the Table 1 would follow one another, e.g. 28, 29, 30, etc.

Results

I miss the title of the Table 2.

Discussion

Figure 7. Drawing of the variants – of what? The title of a figure should exactly tell the content

The authors repeat some of the results in the discussion. They also give some results they did not mention in the “results” section. I miss more clinical orientations of the findings of the study in the discussion.

There are some typing errors in the text.

Author Response

Dear reviewer:

We appreciate your comments, they have made our manuscript improve considerably, below we detail the responses to each of your suggestions.

Reviewer 1:

Abstract

  1. There are abbreviations without explanations in the abstract.

Dear reviewer, all abbreviations have been included in an orderly and coherent manner in the abstract

  1. The summary is too long. You should give only essential information.

The abstract was reduced from 487 words to 394, to be more precise with the information as indicated by the reviewer

Results

  1. The 38 included studies showing recurrent laryngeal nerve variants included in this review were 41, with a cumulative N of 29218.« What does 29218 represent? No of cases included?

We did not find 38 studies, in Table 2 there are 36 included, but in our original document we have 41, for which we added the missing ones to the final document, with respect to the accumulated n we specified what it refers to on the manuscript.

Methodology

  1. Figure 1: I have the impression that the authors have forgotten to put a total number of records in the diagram.

The total number of records was added to the diagram

  1. Table 1. Is it really necessary to show responses in all domains for all included studies? I think it would suffice to say that studies had to meet such or such criteria to be included in the analysis. I would expect the reference numbers in Table 1 to follow each other, e.g. 28, 29, 30, etc.

We think that is necessary to show the details of the AQUA domains for the transparency of the review. We also fixed all numbers in successive order on table 1.

Results

  1. I miss the title of Table 2.

Added title to table 2

Discusion

  1. Figure 7. Drawing of the variants – of what? The title of a figure must say exactly the content.

The characteristics of figure 7 have been detailed

  1. The authors repeat some of the results in the discussion. They also give some results that they didn't mention in the "results" section. I miss more clinical directions from the study findings in the discussion.

The changes suggested by the reviewer have been added, as well as the changes related to the discussion to give it a clinical focus.

Reviewer 2 Report

Thank you for your interesting paper. You put a lot of hard work into your publication! For me, the practical use remains somewhat unclear. Of course, it is useful to know the different variants, at least in theory. During an operation, however, it is necessary to visualize the course of the nerve in the individual patient in order to minimize the risk of damage. I propose to include illustrations of the different types of nerve pathways in relation to the blood vessels and other anatomical structures for clarity.

Author Response

Reviewer 2

Dear reviewers,

We appreciate your comments, they have made our manuscript improve considerably, below we detail the responses to each of your suggestions.

Reviewer 2

  1. Thank you for your interesting paper. You put a lot of hard work into your publication! For me, the practical use remains somewhat unclear. Of course, it is useful to know the different variants, at least in theory. During an operation, however, it is necessary to visualize the course of the nerve in the individual patient in order to minimize the risk of damage. I propose to include illustrations of the different types of nerve pathways in relation to the blood vessels and other anatomical structures for clarity.

Thank you for your kind comment and for the proposition, we added images that help us to illustrate the practical use of this review.

Reviewer 3 Report

The manuscript aims to sum up the knowledge about the RLN. There are several mistakes within the manuscript, however I can see some potential in it. Apart from the technicalities listed below, it would be beneficial for the manuscript if the authors stated more clearly what are the benefits of this study for the clinicians. For example – a commentary on how the prevalence of certain types of RLN could be utilized in the management of the patients would be welcome.

Line 29 – RLN – please explain abbreviations when they first appear in the text

Line 30, 190, 199, 479 and table 2 – „NLR” should be RLN

Line 49 – “variation anatomical” please reconsider using some of the keywords – the results of a search with the aforementioned keywords will probably give poor results. Besides, as I understand, “variation anatomical” was one of the keywords used for the search of articles for this review, but the keywords in this section are supposed to help readers to finds this exact article, after its publication.

Line 135 - to ensure a patent pathway – please explain

Reference numbers are incorrect throughout the manuscript

Table 2, table 3 and table 4 – data in the table is misplaced – the reference numbers are incorrect – check all the data in all the tables!

Lines 312 – 325 – this fragment is almost unreadable due to language mistakes

Table 3 – “Men and % Woman and %” - should be “Women”

Line 341 - Bula et al., 2015 – please be consistent and use one type of references in the text – either the number or the authors’ names and year.

Line 351 - I² - did you mean χ2, or Chi2 ?

Author Response

Dear reviewer:

We appreciate your comments, they have made our manuscript improve considerably, below we detail the responses to each of your suggestions.

  1. Line 29 – RLN – please explain abbreviations when they first appear in the text.

We put the explanations of the abbreviations in their first appearance on the text.

  1. Line 30, 190, 199, 479 and table 2 – „NLR” should be RLN

Thank you, we changed it.

  1. Line 49 – “variation anatomical” please reconsider using some of the keywords – the results of a search with the aforementioned keywords will probably give poor results. Besides, as I understand, “variation anatomical” was one of the keywords used for the search of articles for this review, but the keywords in this section are supposed to help readers to finds this exact article, after its publication.

Thank you, we eliminate it.

  1. Line 135 - to ensure a patent pathway – please explain

We changed it, because of the confusion that it may cause. We meant a permeable pathway.

  1. Reference numbers are incorrect throughout the manuscript

We corrected them.

  1. Table 2, table 3 and table 4 – data in the table is misplaced – the reference numbers are incorrect – check all the data in all the tables!

We correct them.

  1. Lines 312 – 325 – this fragment is almost unreadable due to language mistakes

We rewrote the fragment.

  1. Table 3 – “Men and % Woman and %” - should be “Women

Corrected.

  1. Line 341 - Bula et al., 2015 – please be consistent and use one type of references in the text – either the number or the authors’ names and year.

Thank you, we changed the citation of “Bula et al., 2015”.

  1. Line 351 - I² - did you mean χ2, or Chi2 ?

Chi2

Round 2

Reviewer 1 Report

The authors corrected their paper. I have fund just some minor problems still present in the text.

Lines 128-131: "When the paralysis of the vocal folds is bilateral, the voice decreases considerably, being almost absent, due to the non-movement and narrowness of the vocal folds. Likewise, the vocal folds do not have the capacity to be adducted for phonation, nor abducted to increase ventilation, causing stridor and respiratory obstruction.«

I do not agree that the voice decreases after bilateral vocal fold paralysis. In most cases it is different (high-pitched, weaker) but still present.

The title of the Table 1 is missing.

The titles of Table 3 in Table 4 should be more informative. E.g. Prevalence OF WHAT ? and total number of subjects IN THE INCLUDED STUDIES?

Figure 6: Forest plot of prevalence – OF WHAT?

The authors added a paragraph on clinical applicability of their study (lines 591-530):

One of the most important clinical considerations of the recurrent laryngeal nerve variant is intra-surgical complications, especially associated with the most common surgery in the area that could affect the course of the RLN, the above refers to thyroidectomy and parathyroidectomy which are Preferably performed for treatment of thyroid and parathyroid cancers, patients with this pathological condition present a large amount of volume increase which can be associated with goiter of the thyroid gland and hypervascularization in the area forming collateral branches of the thyroid arteries, a variant of the laryngeal nerve  can be the cause of redistribution in the course of the thyroid arteries or some compression of the same in the lower or upper thyroid region (Figure 8).«

This sentence is 9 lines long. I suggest using shorter sentences in order to be more understandable to the readers.

There are still some typing errors in the text -e.g. line 317, 523, at the end of Table 3.

Did a native English speaker read the text?

Author Response

Dear reviewer:

We appreciate your comments, they have made our manuscript improve considerably, below we detail the responses to each of your suggestions.

  1. Lines 128-131: "When the paralysis of the vocal folds is bilateral, the voice decreases considerably, being almost absent, due to the non-movement and narrowness of the vocal folds. Likewise, the vocal folds do not have the capacity to be adducted for phonation, nor abducted to increase ventilation, causing stridor and respiratory obstruction.«

I do not agree that the voice decreases after bilateral vocal fold paralysis. In most cases it is different (high-pitched, weaker) but still present.

R: We have moved that paragraph and included what the reviewer suggests.

  1. The title of the Table 1 is missing.

R: The title of the table has been included

  1. The titles of Table 3 in Table 4 should be more informative. E.g. Prevalence OF WHAT ? and total number of subjects IN THE INCLUDED STUDIES?

The titles of table 3 and 4 have been modified to be more informative.

  1. Figure 6: Forest plot of prevalence – OF WHAT?

R: Change to: Forest plot of prevalence of RLN variations of the included studies.

  1. The authors added a paragraph on clinical applicability of their study (lines 591-530):

R: Change Applied.

  1. “ One of the most important clinical considerations of the recurrent laryngeal nerve variant is intra-surgical complications, especially associated with the most common surgery in the area that could affect the course of the RLN, the above refers to thyroidectomy and parathyroidectomy which are Preferably performed for treatment of thyroid and parathyroid cancers, patients with this pathological condition present a large amount of volume increase which can be associated with goiter of the thyroid gland and hypervascularization in the area forming collateral branches of the thyroid arteries, a variant of the laryngeal nerve  can be the cause of redistribution in the course of the thyroid arteries or some compression of the same in the lower or upper thyroid region

(Figure 8).«

This sentence is 9 lines long. I suggest using shorter sentences in order to be more understandable to the readers.

R: we have reduced the lines of the sentence mentioned.

  1. There are still some typing errors in the text -e.g. line 317, 523, at the end of Table 3.

R: those bugs have been fixed

  1. Did a native English speaker read the text?

R: The text has been sent to the proofreading service, which has improved the native English and grammar

Reviewer 3 Report

All my commentaries have been satisfactorily replied to. I now consider the study eligible for publication.

Author Response

Dear reviewer:

We appreciate your comments, they have made our manuscript improve considerably, below we detail the responses to each of your suggestions.

All my commentaries have been satisfactorily replied to. I now consider the study eligible for publication.

R: The text has been sent to the proofreading service, which has improved the native English and grammar
